# Kaleidoscope: Learnable Masks for Heterogeneous Multi-agent Reinforcement Learning

**Xinran Li**     **Ling Pan**     **Jun Zhang**

Department of Electronic and Computer Engineering
The Hong Kong University of Science and Technology
`xinran.li@connect.ust.hk, lingpan@ust.hk, eejzhang@ust.hk`

## Abstract

In multi-agent reinforcement learning (MARL), parameter sharing is commonly employed to enhance sample efficiency. However, the popular approach of full parameter sharing often leads to homogeneous policies among agents, potentially limiting the performance benefits that could be derived from policy diversity. To address this critical limitation, we introduce *Kaleidoscope*, a novel adaptive partial parameter sharing scheme that fosters policy heterogeneity while still maintaining high sample efficiency. Specifically, Kaleidoscope maintains one set of common parameters alongside multiple sets of distinct, learnable masks for different agents, dictating the sharing of parameters. It promotes diversity among policy networks by encouraging discrepancy among these masks, without sacrificing the efficiencies of parameter sharing. This design allows Kaleidoscope to dynamically balance high sample efficiency with a broad policy representational capacity, effectively bridging the gap between full parameter sharing and non-parameter sharing across various environments. We further extend Kaleidoscope to critic ensembles in the context of actor-critic algorithms, which could help improve value estimations. Our empirical evaluations across extensive environments, including multi-agent particle environment, multi-agent MuJoCo and StarCraft multi-agent challenge v2, demonstrate the superior performance of Kaleidoscope compared with existing parameter sharing approaches, showcasing its potential for performance enhancement in MARL. The code is publicly available at `https://github.com/LXXXXR/Kaleidoscope`.

## 1   Introduction

Cooperative multi-agent reinforcement learning (MARL) has demonstrated remarkable effectiveness in solving complex real-world decision-making problems across various domains, such as resource allocation (Ying and Dayong, 2005), package delivery (Seuken and Zilberstein, 2007), autonomous driving (Zhou et al., 2021), and robot control (Swamy et al., 2020). To mitigate the challenges posed by the non-stationary and partially observable environments typical of MARL (Yuan et al., 2023), the centralized training with decentralized execution (CTDE) paradigm (Foerster et al., 2016) has become prevalent, inspiring many influential MARL algorithms such as MADDPG (Lowe et al., 2017), COMA (Foerster et al., 2018), MATD3 (Ackermann et al., 2019), QMIX (Rashid et al., 2020), and MAPPO (Yu et al., 2022).

Under the CTDE paradigm, parameter sharing among agents is a commonly adopted practice to improve sample efficiency. However, identical network parameters across agents often lead to homogeneous policies, restricting diversity in behaviors and the overall joint policy representational capacity. This limitation can result in undesired outcomes in certain situations (Christianos et al., 2021; Fu et al., 2022; Kim and Sung, 2023), as shown in Figure 1, impeding further performance

38th Conference on Neural Information Processing Systems (NeurIPS 2024).

gains. An alternative approach is the non-parameter sharing scheme, where each agent possesses its own unique parameters. Nevertheless, while this method naturally supports heterogeneous policies, it suffers from reduced sample efficiency, leading to significant training costs. This is particularly problematic given the current trend towards increasingly large model sizes, with some scaling to trillions of parameters (Zhao et al., 2023; Achiam et al., 2023). Therefore, it is imperative to develop a parameter sharing strategy that enjoys both high sample efficiency and broad policy representational capacity, potentially achieving significantly enhanced performance. While several efforts (Christianos et al., 2021; Kim and Sung, 2023) have explored partial parameter sharing initiated at the start of training, such initializations can be challenging to design without detailed knowledge of agent-specific environmental transitions or reward functions (Christianos et al., 2021).

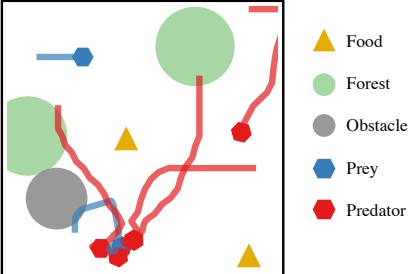

In this work, we build upon insights from previous studies (Christianos et al., 2021; Fu et al., 2022; Kim and Sung, 2023) and introduce *Kaleidoscope*, a novel adaptive partial parameter sharing scheme. It maintains a single set of policy parameters and employs multiple learnable masks to designate the shared parameters. Unlike earlier methods that depend on fixed initializations, Kaleidoscope dynamically learns these masks alongside MARL parameters throughout the training process. This end-to-end training approach inherently integrates environmental information, and its adaptive nature enables Kaleidoscope to dynamically adjust the level of parameter sharing based on the demands of the environment and the learning progress of the agents. The learnable masks facilitate a dynamic balance between full parameter sharing and non-parameter sharing, offering a flexible trade-off between sample efficiency and policy representational capacity

Figure 1: Full parameter sharing confines the policies to be homogeneous. In this example, all predators pursue the same prey, neglecting another prey in the game `World`. Further game details are in Appendix A.2.

through enhanced heterogeneity. Initially, we build Kaleidoscope upon agent networks, where it achieves diverse policies. Following this success, we extend it to multi-agent actor-critic algorithms to encourage heterogeneity among the central critic ensembles for further performance enhancement.

> Just like a *kaleidoscope* uses the reflective properties of rotating mirrors to transform simple shapes into beautiful patterns, our proposed method leverages learnable masks to map a single set of parameters into diverse policies, thereby enhancing task performance.

We summarize our contributions as follows:

- To enable policy heterogeneity among agents for better training flexibility, we adapt the soft threshold reparameterization (STR) technique to learn distinct masks for different agent networks while only maintaining one set of common parameters, effectively balancing between full parameter sharing and non-parameter sharing mechanisms.

- To enhance policy diversity among agents, we introduce a novel regularization term that encourages the pairwise discrepancy between masks. Additionally, we design resetting mechanisms that recycle masked parameters to preserve the representational capacity of the joint networks.

- Through extensive experiments on MARL benchmarks, including multi-agent particle environment (MPE) (Lowe et al., 2017), multi-agent MuJoCo (MAMuJoCo) (Peng et al., 2021) and StarCraft multi-agent challenge v2 (SMACv2) (Ellis et al., 2024), we demonstrate the superior performance of Kaleidoscope over existing parameter sharing approaches.

## 2 Background

**Multi-agent reinforcement learning (MARL)** In MARL, a fully cooperative partially observable multi-agent task is typically formulated as a decentralized partially observable Markov decision process (dec-POMDP) (Oliehoek and Amato, 2016), represented by a tuple $\mathcal{M} = \langle \mathcal{S}, A, P, R, \Omega, O, N, \gamma \rangle$. Here, $N$ denotes the number of agents, and $\gamma \in (0, 1]$ represents the discount factor. At each timestep $t$, with the environment state as $s^t \in \mathcal{S}$, agent $i$ receives a local observation $o_i^t \in \Omega$ drawn from the observation function $O(s^t, i)$ and then follows its local policy

$\pi_i$ to select an action $a_i^t \in A$. Individual actions form a joint action $\boldsymbol{a}^t \in A^N$, leading to a state transition to the next state $s^{t+1} \sim P(s^{t+1}|s^t, \boldsymbol{a}^t)$ and inducing a global reward $r^t = R(s^t, \boldsymbol{a}^t)$. The overall team objective is to learn the joint policies $\boldsymbol{\pi} = \langle \pi_1, \ldots, \pi_N \rangle$ such that the expectation of discounted accumulated reward $G^t = \sum_t \gamma^t r^t$ is maximized.

To learn such policies $\boldsymbol{\pi_\theta}$, various MARL algorithms (Lowe et al., 2017; Foerster et al., 2018; Rashid et al., 2020; Yu et al., 2022) have been developed. For instance, the off-policy actor-critic algorithm MATD3 (Ackermann et al., 2019) serves as an example method. Specifically, the critic networks are updated by minimizing the temporal difference (TD) error loss

$$\mathcal{L}_c(\phi) = \mathbb{E}_{(s^t, \boldsymbol{o}^t, \boldsymbol{a}^t, r^t, s^{t+1}, \boldsymbol{o}^{t+1}) \sim \mathcal{D}} \left[ \left( y^t - Q(s^t, \boldsymbol{a}^t; \phi) \right)^2 \right], \tag{1}$$

with

$$y^t = r^t + \gamma \min_{j=1,2} Q(s^{t+1}, \pi_1(o_1^{t+1}; \theta_1') + \epsilon, \ldots, \pi_N(o_N^{t+1}; \theta_N') + \epsilon; \phi_j), \tag{2}$$

where $\phi$ are the parameters for critics, $\theta$ are the parameters for actor policies and $\theta'$ are the parameters for target actor policies. And $\epsilon$ is the clipped Gaussian noise, given as $\text{clip}(\mathcal{N}(0, \sigma), -c, c)$.

The policy is updated by the deterministic policy gradient algorithm (Silver et al., 2014)

$$\nabla \mathcal{J}(\theta_i) = \mathbb{E}_{(s^t, \boldsymbol{o}^t, \boldsymbol{a}^t, r^t, s^{t+1}, \boldsymbol{o}^{t+1}) \sim \mathcal{D}} \left[ \nabla_{\theta_i} \pi_i(o_i^t; \theta_i) \nabla_{a_i} Q(s^t, a_1, \ldots, a_N|_{a_i = \pi_i(o_i^t; \theta_i)}; \phi_1) \right]. \tag{3}$$

**Soft threshold reparameterization (STR)** Originally introduced in the context of model sparsification, STR (Kusupati et al., 2020) is an unstructured pruning method that achieves notable performance without requiring a predetermined sparsity level. Specifically, STR applies a transformation to the original parameters $W$ as follows

$$\mathcal{S}_g(W, s) = \text{sign}(W) \cdot \text{ReLU}\left( |W| - g(s) \right), \tag{4}$$

where $s$ is a learnable parameter, $\alpha = g(s)$ serves as the pruning threshold, and $\text{ReLU}(\cdot) = \max(\cdot, 0)$. The original supervised learning problem modeled by

$$\min_{\boldsymbol{W}} \mathcal{L}(\boldsymbol{W}; \mathcal{D}) \tag{5}$$

with $\mathcal{D}$ as the data is now transferred to

$$\min_{\boldsymbol{W}, \boldsymbol{s}} \mathcal{L}(\mathcal{S}_g(\boldsymbol{W}, \boldsymbol{s}); \mathcal{D}). \tag{6}$$

Overall, this approach optimizes the learnable pruning threshold alongside the model parameters, facilitating dynamic adjustment to the sparsity level during training.

## 3 Learnable Masks for Heterogenous MARL

In this section, we propose using learnable masks as a low-cost method to enable network heterogeneity in MARL. The core concept, illustrated in Figure 2, is to learn a single set of shared parameters complemented by multiple masks for distinct agents, specifying which parameters to share.

Specifically, in Section 3.1, we first adapt STR into a dynamic partially parameter sharing method, unlocking the joint policy network's capability to represent diverse policies among agents. In Section 3.2, we actively foster policy heterogeneity through a novel regularization term based on the masks. Given that the masking technique could excessively sparsify the network, potentially diminishing its representational capacity, in Section 3.3, we propose a straightforward remedy to periodically reset the parameters based on the outcomes of masking, which additionally mitigates primacy bias. Finally, in Section 3.4, we explore how to further extend this approach within the critic components of actor-critic algorithms to improve value estimations in MARL and further boost performance.

For the sake of clarity, we integrate the proposed Kaleidoscope with the MATD3 (Ackermann et al., 2019) algorithm to demonstrate the concept within this section. Nevertheless, as a versatile partial parameter-sharing technique, our method can readily be adapted to other MARL algorithms. We defer its integration with other MARL frameworks to Appendix A.1.2 and will evaluate them empirically in Section 4.

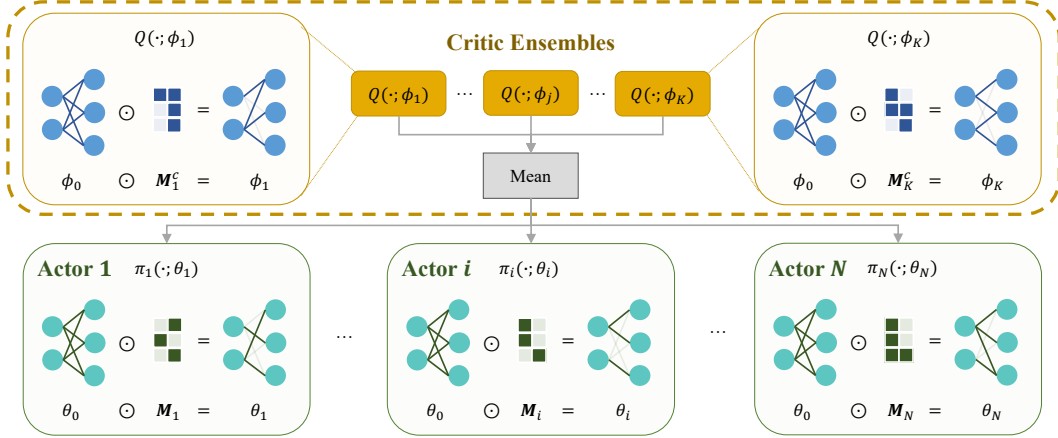

Figure 2: Overall network architecture of Kaleidoscope. It maintains one set of parameters $\theta_0$ with $N$ sets of masks $[M_i]_{i=1}^N$ for actor networks, and one set of parameters $\phi_0$ with $K$ sets of masks $\left[M_j^c\right]_{j=1}^K$ for critic ensemble networks, where $N$ is the number of agents, $K$ is the number of ensembles, and $\odot$ denotes the Hadamard product.

## 3.1 Adaptive partial parameter sharing ♦

The core idea of this work is to learn distinct binary masks $M_i$ for different agents to facilitate differentiated policies, ultimately aiming to improve MARL performance. To achieve this, we apply the STR (Kusupati et al., 2020) technique to the policy parameters with different thresholds dedicated to each agent:

$$\theta_i = \theta_0 \odot M_i, \tag{7}$$

where $\theta_i$ parameterizes the policy for agent $i$, $\theta_0$ is the set of learnable parameters shared by all agents, and $M_i$ is the learnable mask for agent $i$. Specifically, assume $\theta_0 = \left[\theta_0^{(1)}, \ldots, \theta_0^{(N_a)}\right]$, $\theta_i = \left[\theta_i^{(1)}, \ldots, \theta_i^{(N_a)}\right]$ and $M_i = \left[m_i^{(1)}, \ldots, m_i^{(N_a)}\right]$ with $N_a$ being the total parameter count of an agent's network. In line with STR, we compute each element $m_i^{(k)}$ of $M_i$ as $m_i^{(k)} = \mathbb{1}\left[|\theta_0^{(k)}| > \sigma(s_i^{(k)})\right]$, where $\sigma(\cdot)$ denotes the Sigmoid function.

The benefits of such a combination are summarized as follows:

- **Preservation of original MARL learning objectives:** Unlike most of the methods in pruning literature, which primarily aim to minimize the discrepancies between pruned and unpruned networks in terms of weights, loss, or activations (Hoefler et al., 2021; Menghani, 2023; Deng et al., 2020), STR maintains the original goal of minimizing task-specific loss, aligning directly with our objectives to enhance MARL performance.

- **Flexibility in sparsity:** Many classical pruning methods require predefined per-layer sparsity levels (Evci et al., 2020; Ramanujan et al., 2020). Such requirements can complicate our design, with the goal not to gain extreme sparsity but rather to promote heterogeneity through masking. The STR technique is ideal in our case as it does not require predefining sparsity levels, allowing for adaptive learning of the masks.

- **Enhanced network representational capacity:** Utilizing learnable masks for adaptive partial parameter sharing enhances the network's representational capacity beyond traditional full parameter sharing. In full parameter sharing, agents' joint policies are parameterized as $\pi^{\mathrm{ps}}(\cdot|\theta_0) = \langle \pi_1(\cdot|\theta_0), \ldots, \pi_N(\cdot|\theta_0) \rangle$. In contrast, our proposed adaptive partial parameter sharing mechanism parameterizes the joint policies as $\pi^{\mathrm{Kaleidoscope}}(\cdot|\theta_0, M) = \langle \pi_1(\cdot|\theta_0 \odot M_1), \ldots, \pi_n(\cdot|\theta_0 \odot M_N) \rangle$. In the extreme case where all the values in $M_i$ are 1s, the function set represented by $\pi^{\mathrm{Kaleidoscope}}(\cdot|\theta_0, M)$ degrades to that of $\pi^{\mathrm{ps}}(\cdot|\theta_0)$. In other scenarios, it is a superset of that represented by $\pi^{\mathrm{ps}}(\cdot|\theta_0)$.

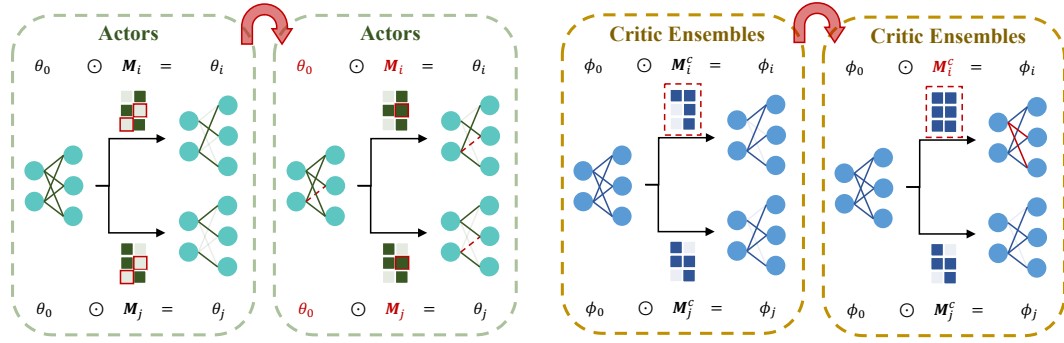

(a) Actors: reinitialize the weights that are masked by all agents with probability $\rho$.

(b) Critic ensembles: reset one set of masks at a time.

Figure 3: Illustration on resetting mechanisms.

## 3.2 Policy diversity regularization ♣

While independently learned masks enable agents to develop distinct policies, without a specific incentive, these policies may still converge to being homogeneous. To this end, we propose to explicitly encourage agent policy heterogeneity by introducing a diversity regularization term maximizing the weighted pairwise distance between network masks, which is defined as

$$\mathcal{J}^{\text{div}}(\boldsymbol{s}) = \sum_{i=1,\ldots,n} \sum_{\substack{j=1,\ldots,n \\ j \neq i}} \|\theta_0 \odot (\boldsymbol{M}_i - \boldsymbol{M}_j)\|_1. \tag{8}$$

This term is inherently non-differentiable due to the indicator function $\mathbb{1}[\cdot]$ inside $\boldsymbol{M}$. To overcome this difficulty, following established practices in the literature (Bengio et al., 2013; Alizadeh et al., 2018), we utilize a surrogate function for gradient approximation:

$$\frac{\partial \mathcal{J}^{\text{div}}}{\partial g(\boldsymbol{s}_i)} = -\tanh\left[\frac{\partial \mathcal{J}^{\text{div}}}{\partial \boldsymbol{M}_i}\right]. \tag{9}$$

We formally provide the overall training objective for actors in Appendix A.1.1.

## 3.3 Periodically reset ♠

As the training with masks proceeds, we observe an increasing sparsity in each agent's network, potentially reducing the overall network capacity. To remedy the issue, we propose a simple approach to periodically reset the parameters that are consistently masked across all $\boldsymbol{M}_i$ with a certain probability $\rho$, which is illustrated in Figure 3a. At intervals defined by $t \mod \text{reset\_interval} == 0$, if the parameter index $k$ satisfies $\forall i, m_i^{(k)} == 0$, we apply the following resetting rule

$$\theta_0^{(k)}, s_1^{(k)}, \ldots, s_N^{(k)} \leftarrow \begin{cases} \text{Reinitialize}[\theta_0^{(k)}, s_1^{(k)}, \ldots, s_N^{(k)}] & \text{with probability } \rho \\ \theta_0^{(k)}, s_1^{(k)}, \ldots, s_N^{(k)} & \text{with probability } 1 - \rho \end{cases}. \tag{10}$$

This resetting mechanism recycles the weights masked as zeros by all the masks, preventing the networks from becoming overly sparse. A side benefit of this resetting mechanism is the enhancement of neural plasticity (Lyle et al., 2023; Nikishin et al., 2024), which helps alleviate the primacy bias (Nikishin et al., 2022) in reinforcement learning. Unlike methods that reinitialize entire layers resulting in abrupt performance drops (Nikishin et al., 2022), our resetting approach selectively targets weights as indicated by the learnable masks, thus avoiding significant performance disruptions, as shown in Section 4.

## 3.4 Critic ensembles with learnable masks

In actor-critic algorithm frameworks, we further apply Kaleidoscope to central critics as an efficient way to implement ensemble-like critics. By facilitating dynamic partial parameter sharing, Kaleidoscope enables heterogeneity among critic ensembles. Furthermore, by regularizing the diversity

among critic functions, we can control ensemble variances. This approach is elaborated in subsequent paragraphs.

♦ **Adaptive partial parameter sharing for critic ensembles**  In the standard MATD3 algorithm (Ackermann et al., 2019), two critics with independent parameters are maintained to mitigate overestimation risks. However, using separate parameters typically results in a low update-to-data (UTD) ratio (Hiraoka et al., 2022). To address this issue, we propose to enhance the UTD ratio by employing Kaleidoscope parameter sharing among ensembles of critics. Specifically, we maintain a single set of parameters $\phi_0$ and $K$ masks $\left[\boldsymbol{M}_j^c\right]_{j=1}^K$ to distinguish the critic functions, resulting in $K$ ensembles $[Q(\cdot; \phi_j)]_{j=1}^K$ with $\phi_j = \phi_0 \odot \boldsymbol{M}_j^c$.

To be specific, we update the critic networks by minimizing the temporal difference (TD) error loss

$$\mathcal{L}_c(\phi_j) = \mathbb{E}_{(s^t, \boldsymbol{a}^t, s_{t+1}) \sim \mathcal{D}} \left[ \left( y^t - Q(s^t, \boldsymbol{a}^t; \phi_j) \right)^2 \right], \tag{11}$$

with

$$y^t = r^t + \gamma \min_{j=1,\ldots,K} Q(s^{t+1}, \pi_1(o_1^{t+1}; \theta_1') + \epsilon, \ldots, \pi_n(o_N^{t+1}; \theta_N') + \epsilon; \phi_j). \tag{12}$$

And the policies are updated by the mean estimation of the ensembles as

$$\nabla \mathcal{J}(\theta_i) = \mathbb{E}_{s^t \sim \mathcal{D}} \left[ \nabla_{\theta_i} \pi_i(o_i^t; \theta_i) \nabla_{a_i} \frac{1}{K} \sum_{j=1}^K \left[ Q(s^t, a_1, \ldots, a_N|_{a_i = \pi_i(o_i^t; \theta_i)}; \phi_j) \right] \right]. \tag{13}$$

♣ **Critic ensembles diversity regularization**  As in Section 3.2, we also apply diversity regularization to critic masks to prevent critics functions from collapsing to identical ones. The diversity regularization to maximize for the critic ensembles is expressed as

$$\mathcal{J}_c^{\text{div}}(\boldsymbol{s}^c) = \sum_{i=1,\ldots,K} \sum_{\substack{j=1,\ldots,K \\ j \neq i}} \|\phi_0 \odot (\boldsymbol{M}_i^c - \boldsymbol{M}_j^c)\|_1. \tag{14}$$

Intuitively, as training progresses, this term encourages divergence among the critic masks, leading to increased model estimation uncertainty. This process fosters a gradual shift from overestimation to underestimation. As discussed in prior research (Hiraoka et al., 2022; Lan et al., 2020; Chen et al., 2021; Wang et al., 2021b), overestimation can encourage exploration, beneficial in early training stages, whereas underestimation alleviates error accumulation (Fujimoto et al., 2018), which is preferred in the late training stage. We formally provide the overall training objective for critic ensembles in Appendix A.1.1.

♠ **Periodically reset**  To further promote diversity among critic ensembles and counteract the reduction in network capacity caused by masking, we implement a resetting mechanism similar to that described in Section 3.3. In particular, we sequentially reinitialize the masks $\boldsymbol{M}_j^c$ following a cyclic pattern, as illustrated in Figure 3b. In this way, each critic function's mask is trained on distinct data segments, leading to different biases.

In summary, by adopting Kaleidoscope parameter sharing with learnable masks, we establish a cost-effective implementation for critic ensembles that enjoy a high UTD ratio. Through enforcing distinctiveness among the masks, we subtly control the differences among critic functions, thereby improving the value estimations in MARL.

## 4   Experimental Results

In this section, we integrate Kaleidoscope with the value-based MARL algorithm QMIX and the actor-critic MARL algorithm MATD3, and evaluate them across eleven scenarios in three benchmark tasks.

Table 1: Methods compared in the experiments. Here, "adaptive" indicates whether the sharing scheme evolves during training.

| Methods | Paradigm | Sharing level | Adaptive | Descriptions |
|---------|----------|---------------|----------|--------------|
| NoPS | No sharing | - | No | Agents have distinct parameters |
| FuPS | Full sharing | Networks | No | Agents share all the parameters |
| FuPS + ID | Full sharing | Networks | No | Agents share all the parameters with agent IDs in input |
| SePS | Partial sharing | Networks | No | Agents are clustered to share parameters within each cluster |
| MultiH | Partial sharing | Layers | No | Agents share all the parameters except for distinct action heads |
| SNP | Partial sharing | Neurons | No | Agents share specific neurons based on fixed, random pruning |
| Kaleidoscope | Partial sharing | Weights | Yes | Agents share parameters based on distinct, learnable masks |

## 4.1 Experimental Setups

**Environment descriptions**   We test our proposed Kaleidoscope on three benchmark tasks: MPE (Lowe et al., 2017), MaMuJoCo (Peng et al., 2021) and SMACv2 (Ellis et al., 2024). For the discrete tasks MPE and SMACv2, we integrate Kaleidoscope and baselines with QMIX (Rashid et al., 2020) and assess the performance.  For the continuous task MaMuJoCo, we employ MATD3 (Ackermann et al., 2019). We use five random seeds for MPE and MaMuJoCo and three random seeds for SMACv2, reporting averaged results and displaying the $95\%$ confidence interval with shaded areas.  The chosen benchmark tasks reflect a mix of discrete and continuous action spaces and both homogeneous and heterogeneous agent types, detailed further in Appendix A.2.

**Baselines**   In the following, we compare our proposed Kaleidoscope with baselines (Christianos et al., 2021; Kim and Sung, 2023), as listed in Table 1. For both Kaleidoscope and the baselines, in scenarios with fixed agent types (MPE and MaMuJoCo), we assign one mask per agent. For SMACv2, where agent types vary, we assign one mask per agent type. We use official implementations of the baselines where available; otherwise, we closely follow the descriptions from their respective papers, integrating them into QMIX or MATD3. Hyperparameters and further details are provided in Appendix A.1.3.

## 4.2 Results

**Performance**   We present the comparative performance of Kaleidoscope and baselines in Figure 4 and Figure 5. Overall, Kaleidoscope demonstrates superior performance, attributable to the flexibility of the learnable masks and the effectiveness of diversity regularization. Additionally, we observe that FuPS + ID generally outperforms NoPS, except for the `Ant-v2-4x2` scenario (Figure 4c). This advantage is largely due to FuPS's higher sample efficiency; a single transition data sample updates the model parameters $N$ times in FuPS + ID, once for each agent, compared to just once in NoPS. Consequently, FuPS + ID models learn faster from the same number of transitions. Similarly, Kaleidoscope benefits from this mechanism as it shares weights among agents, allowing a single transition to update the model parameters multiple times.  Furthermore, by integrating policy heterogeneity through learnable masks, Kaleidoscope enables diverse agent behaviors, as illustrated in the visualization results in Figure 8. Ultimately, Kaleidoscope effectively balances parameter sharing and diversity, outperforming both full parameter sharing and non-parameter sharing approaches.

**Cost analysis**   Despite its superior performance, Kaleidoscope does not increase computational complexity at test time compared to the baselines. We report the test time averaged FLOPs comparison of Kaleidoscope and baselines in Table 2. We see that due to the masking technique, Kaleidoscope has lower FLOPs compared to baselines, thereby enjoying a faster inference speed when being deployed.

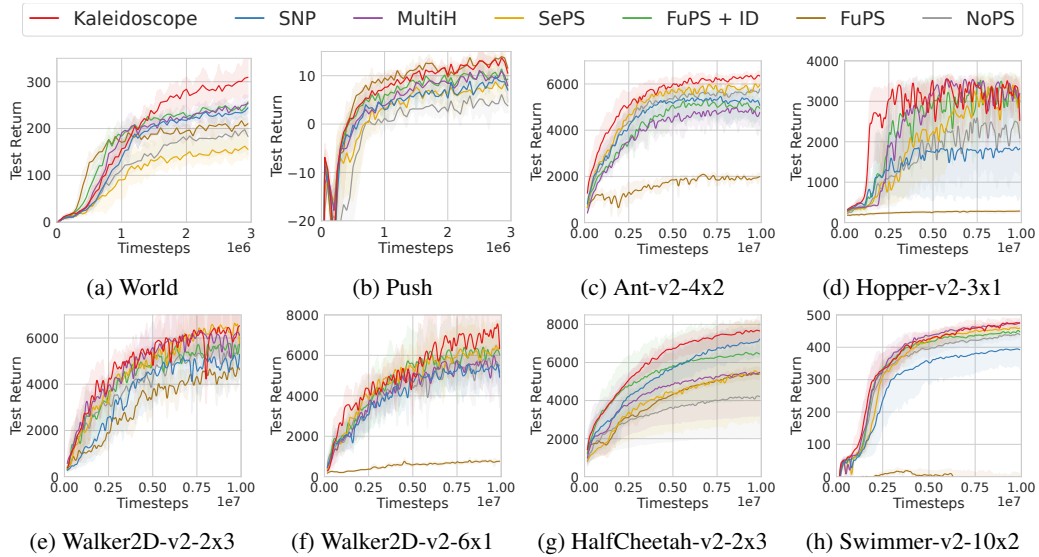

Figure 4: Performance comparison with baselines on MPE and MaMuJoCo benchmarks.

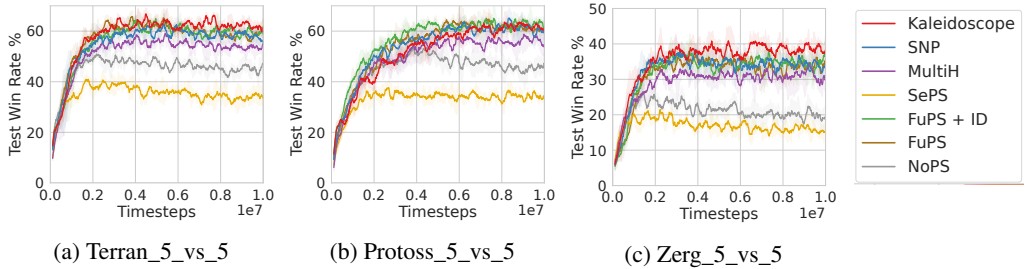

Figure 5: Performance comparison with baselines on SMACv2 benchmarks.

**Ablation studies** We conduct ablation studies to assess the impact of key components in Kaleidoscope, with results presented in Figure 6. Specifically, we compare Kaleidoscope with three ablations: 1) *Kaleidoscope w/o reg*, which lacks the regularization term in Equation (8) that encourages the masks to be distinct. 2) *Kaleidoscope w/o reset*, which does not reset parameters. 3) *Kaleidoscope w/o ce*, which does not use Kaleidoscope parameter sharing in critic ensembles and instead maintains two independent sets of parameters for critics. From the results, we observe that diversity regularization contributes the most to the performance of Kaleidoscope. Without it, masking degrades the performance due to the reduced number of parameters in each policy network. Resetting primarily aids learning in the late stages of training when needed, which aligns with the observation made by Nikishin et al. (2022). Notably, even with resetting, the performance does not experience abrupt drops thanks to the guidance provided by the masks on where to reset. When ablating the critic ensembles with Kaleidoscope parameter sharing, we observe inferior performance from the beginning of the training. This is because the critic ensembles with Kaleidoscope parameter sharing enable a higher UTD ratio of the critics, as discussed in Section 3.4.

Furthermore, we conduct experiments to study the impact of mask designs. The results are shown in Figure Figure 7. Specifically, we compare original Kaleidoscope with two alternative mask design choices: 1) *Kaleidoscope w/ neuron masks*, where adaptive masking techniques are applied to neurons rather than weights. 2) *Kaleidoscope w/ fixed masks*, where the masks are initialized at the beginning of training and kept fixed throughout the learning process. The results show that performance drops with either alternative design choice, demonstrating that Kaleidoscope's superior performance originates from the flexibility of the learnable masks on weights.

More results on hyperparameter analysis are included in Appendix B.2.

Table 2: Averaged FLOPs (with calculation methods detailed in Appendix A.3) across different methods. Results are first normalized with respect to the FuPS + ID model for each scenario and then averaged across scenarios within each environment (detailed results in Appendix B.1). The lowest costs are highlighted in **bold**.

| Methods | NoPS | FuPS | FuPS +ID | SePS | MultiH | SNP | Kaleidoscope |
|---------|------|------|----------|------|--------|-----|--------------|
| MPE | 1.0x | 0.992x | 1.0x | 1.0x | 1.0x | 0.988x | **0.901x** |
| MaMuJoCo | 1.0x | 0.985x | 1.0x | 1.0x | 1.0x | 0.900x | **0.680x** |
| SMACv2 | 1.0x | 0.992x | 1.0x | 1.0x | 1.0x | 0.988x | **0.890x** |

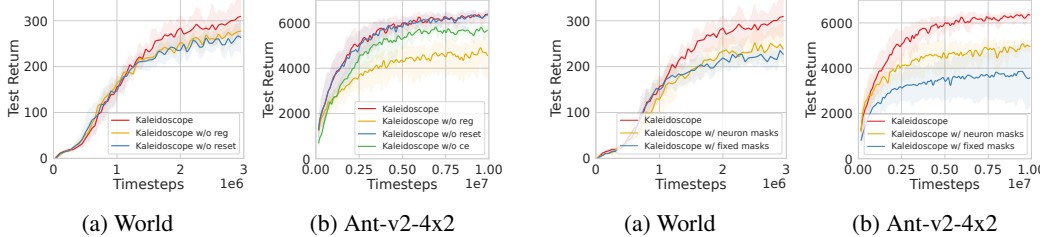

| (a) World | (b) Ant-v2-4x2 | (a) World | (b) Ant-v2-4x2 |
|-----------|----------------|-----------|----------------|

Figure 6: Ablation studies.    Figure 7: Comparison on mask designs.

**Visualization** We visualize the trained policies of Kaleidoscope on `World`, as shown in Figure 8a. The agents exhibit cooperative divide-and-conquer strategies (four red agents divide into two teams and surround the preys), contrasting with the homogeneous policies depicted in Figure 1. We further examine the distinctions in the agents' masks and present the results in Figure 8b. First, we observe that by the end of the training, each agent has developed a unique mask, revealing that distinct masks facilitate diverse policies by selectively activating different segments of the neural network weights. Second, throughout the training process, we note that the differences among the agents' masks evolve dynamically. This observation confirms that Kaleidoscope effectively enables dynamic parameter sharing among the agents based on the learning progress, empowered by the adaptability of the learnable masks. More visualization results are provided in Appendix B.3.

## 5   Related Work

**Parameter sharing** First introduced by Tan (1993), parameter sharing has been widely adopted in MARL algorithms (Foerster et al., 2018; Rashid et al., 2020; Yu et al., 2022), due to its simplicity and high sample efficiency (Grammel et al., 2020). However, schemes without parameter sharing typically offer greater flexibility for policy representation. To balance sample efficiency with policy representational capacity, some research efforts aim to find effective partial parameter sharing schemes. Notably, SePS (Christianos et al., 2021) first clusters agents based on their transitions at the start of training and restricts parameter sharing within these clusters. Subsequently, SNP (Kim and Sung, 2023) enables partial parameter sharing by utilizing the lottery ticket hypothesis (Su et al., 2020) to initialize heterogeneous network structures. Concurrent to our work, AdaPS (Li et al., 2024) combines SNP and SePS by proposing a cluster-based partial parameter sharing scheme. While these methods have shown promise in certain domains, their performance potential is often limited by the static nature of the parameter sharing schemes set early in training. Our proposed Kaleidoscope distinguishes itself by dynamically learning specific parameter sharing configurations alongside the development of MARL policies, thereby offering enhanced training flexibility.

**Agent heterogeneity in MARL** To incorporate agent heterogeneity in MARL and enable diverse behaviors among agents, previous methods have explored concepts such as diversity and roles. Specifically, diversity-based approaches aim to enhance pairwise distinguishability among agents based on identities (Jiang and Lu, 2021), trajectories (Li et al., 2021), or credits assignment (Liu et al., 2023; Hu et al., 2023) through contrastive learning techniques. Concurrently, role-based strategies, sometimes referred to as skills (Yang et al., 2020) or subtasks (Yuan et al., 2022), employ conditional policies to differentiate agents by assigning them to various conditions. These conditions may be based on agent identities (Yang et al., 2022), local observations (Yang et al., 2020), local

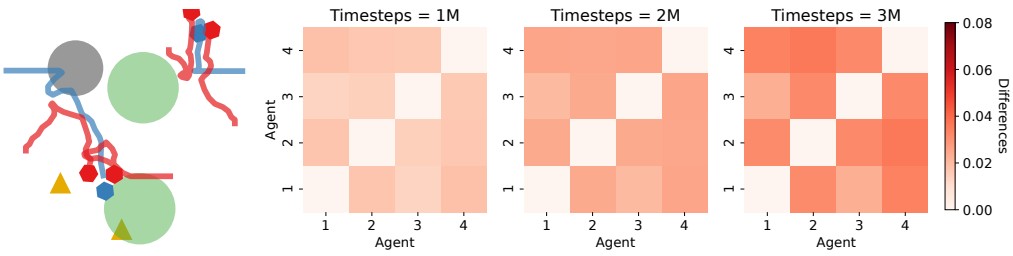

(a) Trained policies.    (b) Pairwise mask differences among agents at different training timesteps.

Figure 8: Visualization on `World`.

histories (Wang et al., 2020, 2021a; Yuan et al., 2022) or joint histories (Liu et al., 2021; Iqbal et al., 2022; Zeng et al., 2023). This line of researches mainly focus on module design and operate separately from parameter-level adjustments, making them orthogonal to our approach. Nevertheless, integrating these methods with our work could potentially enhance performance further.

**Sparse networks in deep reinforcement learning (RL)**    Although relatively few, there are some noteworthy recent attempts to find sparse networks for deep RL. In particular, PoPS (Livne and Cohen, 2020) prunes the dense networks post-training, achieving significantly reduced execution time complexity. Additionally, (Yu et al., 2020) validate the lottery ticket hypothesis within the RL domain, producing high-performance models even under extreme pruning rates. Subsequent efforts, including DST (Sokar et al., 2022), TE-RL* (Graesser et al., 2022) and RLx2 (Tan et al., 2023) employ topology evolution (TE) techniques to further decrease the training costs. While these developments utilize sparse training techniques, which are similar to the methods we employ, their primary focus is on reducing training and execution costs in single-agent settings. In contrast, our work leverages sparse network strategies as a means to enhance parameter sharing techniques, aiming to improve MARL performance.

## 6    Conclusions and Future Work

In this work, we introduced *Kaleidoscope*, a novel adaptive partial parameter sharing mechanism for MARL. It leverages distinct learnable masks to facilitate network heterogeneity, applicable to both agent policies and critic ensembles. Specifically, Kaleidoscope is built on three technical components: STR-empowered learnable masks, network diversity regularization, and a periodic resetting mechanism. When applied to agent policy networks, Kaleidoscope balances sample efficiency and network representational capacities. In the context of critic ensembles, it improves value estimations. By combining our proposed Kaleidoscope with QMIX and MATD3, we have empirically demonstrated its effectiveness across various MARL benchmarks. This study shows great promises in developing adaptive partial parameter sharing mechanisms to enhance the performance of MARL. For future work, it is interesting to further extend Kaleidoscope to other domains such as offline MARL or meta-RL.

## Acknowledgements

This work was supported by the Hong Kong Research Grants Council under the NSFC/RGC Collaborative Research Scheme grant CRS_HKUST603/22. And we thank the anonymous reviewers for their valuable feedback and suggestions.

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

# A Experimental details

## A.1 Implementation details

### A.1.1 Kaleidoscope with MATD3

**Critic ensembles** When incorporating Kaleidoscope into the MATD3 algorithm, the overall training loss for the critic ensembles becomes:

$$\mathcal{L}_c^{\text{all}}(\phi_0, \boldsymbol{s}^c) = \sum_{j=1,...,K} \mathcal{L}_c^{\text{all}}(\phi_j) = \sum_{j=1,...,K} \mathcal{L}_c(\phi_0, \boldsymbol{s}_j^c) - \alpha^d \cdot \mathcal{J}_c^{\text{div}}(\boldsymbol{s}^c), \tag{15}$$

with $\mathcal{L}_c(\phi_0, \boldsymbol{s}_j^c)$ being the original MATD3 loss given in Equation (11), $\mathcal{J}_c^{\text{div}}(\boldsymbol{s}^c)$ being the diversity regularization given in Equation (14), and $\alpha^d$ being a coefficient balancing the original MARL objective and the proposed diversity regularization. Note that although $\mathcal{J}_c^{\text{div}}(\boldsymbol{s}^c)$ contains parameters $\phi_0$, we stop the gradients for $\phi_0$ in $\mathcal{J}_c^{\text{div}}(\boldsymbol{s}^c)$.

In the implementation, we apply layer-wise weights to the diversity regularization term $\mathcal{J}_c^{\text{div}}(\boldsymbol{s}^c)$, which is defined as

$$\mathcal{J}_c^{\text{div}}(\boldsymbol{s}^c) = \sum_{l=1,...,L} w_l \cdot \sum_{i=1,...,K} \sum_{\substack{j=1,...,K \\ j \neq i}} \|\phi_0 \odot (\boldsymbol{M}_{i,l}^c - \boldsymbol{M}_{j,l}^c)\|_1, \tag{16}$$

where $l$ denotes the layer index of the neuron networks, $L$ represents the total number of layers, $\boldsymbol{M}_{i,l}^c$ is the mask for agent $i$ at layer $l$, and the layer-wise weights are set as $w_l = 2^l$. The intuition behind this choice is that features closer to the output tend to be more compact (Kusupati et al., 2020); consequently, assigning larger regularization weights to these layers may have a more significant impact on the output action decisions. Our initial experiments empirically demonstrate that setting $w_l = 2^l$ improves performance compared to the case where $w_l = 1$. Based on these findings, we maintain this design choice throughout all our experiments.

In practice, we adaptively adjust $\alpha^d$ while maintaining a constant ratio between the MATD3 loss and the diversity loss, which is treated as a hyperparameter:

$$\alpha^d = \frac{|\sum_{j=1,...,K} \mathcal{L}_c(\phi_0, \boldsymbol{s}_j^c)|}{|\mathcal{J}_c^{\text{div}}(\boldsymbol{s}^c)|} \cdot \alpha, \tag{17}$$

where $\alpha$ is a hyperparameter, and the gradients for $\frac{|\sum_{j=1,...,K} \mathcal{L}_c(\phi_0, \boldsymbol{s}_j^c)|}{|\mathcal{J}_c^{\text{div}}(\boldsymbol{s}^c)|}$ are stopped.

**Actors** For the actors, the training objective is to maximize the following term

$$\mathcal{J}^{\text{all}}(\theta_0, \boldsymbol{s}) = \sum_{i=1,...,n} \mathcal{J}^{\text{all}}(\theta_i) = \sum_{i=1,...,n} \mathcal{J}(\theta_0, \boldsymbol{s}_i) + \beta^d \cdot \mathcal{J}^{\text{div}}(\boldsymbol{s}), \tag{18}$$

where $\mathcal{J}(\theta_0, \boldsymbol{s}_i)$ is the original actor objective defined in Equation (3), $\mathcal{J}^{\text{div}}(\boldsymbol{s})$ is the diversity regularization given in Equation (8) with layer-wise weights as in Equation (16), $\beta^d$ is the regularization coefficient. The value of $\beta^d$ is determined by

$$\beta^d = \frac{|\sum_{i=1,...,n} \mathcal{J}(\theta_0, \boldsymbol{s}_i)|}{|\mathcal{J}^{\text{div}}(\boldsymbol{s})|} \cdot \beta, \tag{19}$$

where $\beta$ is a constant hyperparameter, similar to the approach used in Equation (17) for the critic ensembles.

### A.1.2 Kaleidoscope with QMIX

When incorporating Kaleidoscope into the QMIX algorithm (Rashid et al., 2020), we apply Kaleidoscope parameter sharing only to the local Q networks. Consequently, the training loss is defined as:

$$\mathcal{L}^{\text{all}}(\theta_0, \boldsymbol{s}) = \mathcal{L}(\theta_0, \boldsymbol{s}) - \beta^d \cdot \mathcal{J}^{\text{div}}(\boldsymbol{s}), \tag{20}$$

where

$$\mathcal{L}(\theta_0, \boldsymbol{s}) = \mathbb{E}_{(s^t, \boldsymbol{o}^t, \boldsymbol{a}^t, r^t, s^{t+1}, \boldsymbol{o}^{t+1}) \sim \mathcal{D}} \left[ \left( y^{tot} - Q_{tot}(s^t, \boldsymbol{o}^t, \boldsymbol{a}^t; \theta_0, \boldsymbol{s}) \right)^2 \right], \tag{21}$$

with $y^{tot} = r + \gamma \max_{\boldsymbol{a}} Q_{tot}(s^{t+1}, \boldsymbol{o}^{t+1}, \boldsymbol{a}; \theta^-)$ and $\theta^-$ representing the parameters of a target network as in DQN.

### A.1.3 Network architecture and hyperparameters

**Codebase**   Our implementation of Kaleidoscope and baseline algorithms are based on the following codebase:

- HARL (Zhong et al., 2024) (MATD3 implementation): `https://github.com/PKU-MARL/HARL`
- EPyMARL (Papoudakis et al., 2021) (QMIX implementation for MPE): `https://github.com/uoe-agents/epymarl`
- PyMARL2 (Hu et al., 2021) (QMIX implementation for SMACv2): `https://github.com/benellis3/pymarl2`
- SePS (Christianos et al., 2021): `https://github.com/uoe-agents/seps`

The code for Kaleidoscope is publicly available at `https://github.com/LXXXXR/Kaleidoscope`.

**Network architecture**   In line with prior works (Zhong et al., 2024; Papoudakis et al., 2021; Hu et al., 2021), we employ deep neural networks consisting of multilayer perceptrons (MLPs) with rectified linear unit (ReLU) activation functions and gated recurrent units (GRUs) to parameterize the actor and critic networks. Moreover, when the masking technique is applied to critic ensembles, we incorporate layer normalization between the MLP layers and ReLU activations (Hiraoka et al., 2022). In Kaleidoscope, the masking technique is applied to the MLP layers, and the resetting mechanisms described in Sections 3.3 and 3.4 are applied to the last three layers of the respective neural networks, following Nikishin et al. (2022).

**Hyperparameters**   To ensure a fair comparison, we implement our method and all the baselines using the same codebase with the same set of hyperparameters, with the exception of method-specific ones. The common hyperparameters are listed in Tables 3 to 5. The Kaleidoscope-specific hyperparameters are provided in Table 6.

Table 3: Common hyperparameters used for MATD3 in the MaMuJoCo domain.

| Hyperparameter | Value |
|---|---|
| Number of layers | 3 |
| Hidden sizes | 256 |
| Discount factor $\gamma$ | 0.99 |
| Rollout threads | 10 |
| Critic lr | $1 \times 10^{-3}$ |
| Actor lr | $5 \times 10^{-4}$ |
| Exploration noise | 0.1 |
| Batch size | 1000 |
| Replay buffer size | $1 \times 10^6$ |
| Number of environment steps | $10 \times 10^6$ |
| n_step[1] | $(5, 10, 20)$ |

[1] Here we adopt the per-scenario finetuned value for this hyperparameter as provided by HARL.

### A.2 Environmental details

**Codebase**   The environments used in this work are listed below with descriptions in Table 7.

- MaMuJoCo (Peng et al., 2021): `https://github.com/schroederdewitt/multiagent_mujoco`
- MPE (Lowe et al., 2017; Papoudakis et al., 2021): `https://github.com/semitable/multiagent-particle-envs`
- SMACv2 (Ellis et al., 2024): `https://github.com/oxwhirl/smacv2`

Table 4: Common hyperparameters used for QMIX in the MPE domain.

| Hyperparameter | Value |
|---|---|
| Number of layers | 5 |
| Hidden sizes | 64 |
| Discount factor $\gamma$ | 0.99 |
| Lr | $5 \times 10^{-4}$ |
| Initial $\epsilon$ | 1.0 |
| Final $\epsilon$ | 0.05 |
| Batch size | 32 |
| Replay buffer size | 5000 |
| Number of environment steps | $3 \times 10^{6}$ |
| Double Q | True |

Table 5: Common hyperparameters used for QMIX in the SMACv2 domain.

| Hyperparameter | Value |
|---|---|
| Number of layers | 5 |
| Hidden sizes | 64 |
| Discount factor $\gamma$ | 0.99 |
| Lr | $1 \times 10^{-3}$ |
| Initial $\epsilon$ | 1.0 |
| Final $\epsilon$ | 0.05 |
| Batch size | 128 |
| Replay buffer size | 5000 |
| Number of environment steps | $10 \times 10^{6}$ |
| Double Q | False |

**MPE**    We extend the scenario settings provided in the original codebase to increase the complexity and challenge of the tasks. In `World`, we set the number of predators (agents) to 4, the number of prey to 2, the number of obstacles to 1, and the number of forests to 2. The objective of the game is for the predators to approach the prey while avoiding collisions with obstacles. The prey is attracted to the food and can hide from the predators in the forests. In `Push`, we set the number of agents to 5, the number of adversaries to 2, and the number of landmarks to 2. The goal of the game is for the agents to push the adversaries away from the landmarks. In both scenarios, we pretrain the adversary (prey) policies using the MADDPG algorithm Lowe et al. (2017) and use these pretrained policies to test the performance of different algorithms

### A.3   FLOPs calculation

To calculate the number of floating-point operations (FLOPs) for a single forward pass of a sparse model, we sum the total number of multiplications and additions layer by layer, following the approach in Evci et al. (2020). For a fully-connected layer, the FLOPs are computed as follows:

$$\text{FLOPs} = 2 \times (1 - \text{Sparsity}) \times \text{In} \times \text{Out}. \tag{22}$$

For a GRU cell, the FLOPs are computed as:

$$\text{FLOPs} = 2 \times (3 \times \text{Hidden}^2 + 3 \times \text{In} \times \text{Hidden} + 13 \times \text{Hidden}). \tag{23}$$

### A.4   Experimental Infrastructure

The experiments on the SMACv2 benchmark were conducted using NVIDIA GeForce RTX 3090 GPUs, while the experiments on other benchmarks were performed using NVIDIA GeForce RTX 3080 GPUs. Each experimental run required less than 2 days to complete.

Table 6: Hyperparameters used for Kaleidoscope.

| Hyperparameter | Environment | Value |
|---|---|---|
| Actor diversity coefficient $\beta$ | MaMuJoCo | 0.1 |
| | MPE | 0.5 |
| | SMACv2 | 5.0 |
| Actors reset probability $\rho$ | MaMuJoCo | 0.5 |
| | MPE | 0.1 |
| | SMACv2 | 0.2 |
| Actor reset interval | MaMuJoCo | $1 \times 10^6$ |
| | MPE | $200 \times 10^3$ |
| | SMACv2 | $1 \times 10^6$ |
| Number of critic ensembles $K$ | MaMuJoCo | 5 |
| Critic ensembles diversity coefficient $\alpha$ | MaMuJoCo | 0.1 |
| Critic reset interval | MaMuJoCo | $800 \times 10^3$ |

Table 7: Environments details.

| Environment | Action space | Agent types | Scenarios | Number of agents |
|---|---|---|---|---|
| MaMuJoCo | Continuous | Heterogeneous, fixed | Ant-v2-4x2 | 4 |
| | | | Hopper-v2-3x1 | 3 |
| | | | Walker2D-v2-2x3 | 2 |
| | | | Walker2D-v2-6x1 | 6 |
| | | | HalfCheetah-v2-2x3 | 2 |
| | | | Swimmer-v2-10x2 | 10 |
| MPE | Discrete | Homogeneous, fixed | World | 4 |
| | | | Push | 5 |
| SMACv2 | Discrete | Heterogeneous, dynamic | Terran_5_vs_5 | 5 |
| | | | Protoss_5_vs_5 | 5 |
| | | | Zerg_5_vs_5 | 5 |

# B More results and discussion

## B.1 Detailed Costs

We provide per-scenario FLOPs across different methods in Table 8 as a supplement for Table 2.

Table 8: Averaged FLOPs for different methods. Results are normalized w.r.t. the FuPS + ID model. The lowest costs are highlighted in **bold**.

| Scenarios | NoPS | FuPS | FuPS +ID | SePS | MultiH | SNP | Kaleidoscope |
|---|---|---|---|---|---|---|---|
| World | 1.0x | 0.993x | 1.0x | 1.0x | 1.0x | 0.988x | **0.897x** |
| Push | 1.0x | 0.991x | 1.0x | 1.0x | 1.0x | 0.988x | **0.904x** |
| Ant-v2-4x2 | 1.0x | 0.990x | 1.0x | 1.0x | 1.0x | 0.900x | **0.640x** |
| Hopper-v2-3x1 | 1.0x | 0.989x | 1.0x | 1.0x | 1.0x | 0.900x | **0.721x** |
| Walker2D-v2-2x3 | 1.0x | 0.992x | 1.0x | 1.0x | 1.0x | 0.900x | **0.731x** |
| Walker2D-v2-6x1 | 1.0x | 0.979x | 1.0x | 1.0x | 1.0x | 0.900x | **0.763x** |
| HalfCheetah-v2-2x3 | 1.0x | 0.993x | 1.0x | 1.0x | 1.0x | 0.900x | **0.614x** |
| Swimmer-v2-10x2 | 1.0x | 0.968x | 1.0x | 1.0x | 1.0x | 0.900x | **0.611x** |
| Terran_5_vs_5 | 1.0x | 0.992x | 1.0x | 1.0x | 1.0x | 0.988x | **0.890x** |
| Protoss_5_vs_5 | 1.0x | 0.992x | 1.0x | 1.0x | 1.0x | 0.988x | **0.895x** |
| Zerg_5_vs_5 | 1.0x | 0.992x | 1.0x | 1.0x | 1.0x | 0.988x | **0.885x** |

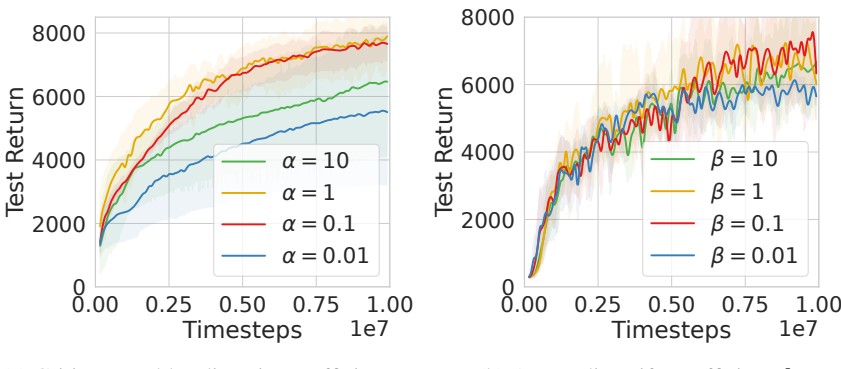

(a) Critic ensembles diversity coefficient $\alpha$.   (b) Actors diversify coefficient $\beta$.

Figure 9: Hyperparameter analysis.

## B.2   Hyperparameter Analysis

We conduct further analysis on the hyperparameters $\alpha$ and $\beta$, and present the results in Figure 9. The hyperparameter $\alpha$ controls the variance of the critic ensembles. As shown in Figure 9a, we observe that an excessively small $\alpha$ results in degraded performance because it reduces the critic ensembles to a single critic network, causing the value estimation to suffer from severe overestimation. Conversely, an excessively large $\alpha$ also deteriorates performance, possibly due to increased estimation bias. For $\beta$, as illustrated in Figure 9b, an overly small $\beta$ leads to degraded performance because it reduces the Kaleidoscope parameter sharing to full parameter sharing, confining the policies to be identical. An overly large $\beta$ also negatively impacts performance, as it may cause the training objective to deviate too much from minimizing the original MARL loss.

In general, we recommend setting both hyperparameters between $0.1$ and $1$. However, the optimal hyperparameter values may vary across different scenarios. For fair comparisons, we maintain the same set of hyperparameters across all scenarios in our experiments. Nevertheless, further tuning of these hyperparameters has the potential to enhance performance.

## B.3   Further Visualization Results

To better understand how learnable masks in Kaleidoscope affect the performance through policies, we visualize the pairwise mask differences among agents and the agent trajectories at different training stages in Figure 10. As training progresses, the test return increases and diversity loss decreases, indicating better performance and greater diversity among agent policies. Correspondingly, mask differences among agents increase, and the agent trajectory distribution becomes more diverse.

## B.4   Limitations

Here we discuss some limitations of Kaleidoscope.

First, as suggested by results in Appendix B.2, the optimal hyperparameters vary from scenario to scenario. Therefore, using the same hyperparameters across all scenarios may not yield the best performance for Kaleidoscope. Developing an automatic scheme that utilizes environmental information to determine these hyperparameters would be beneficial.

Second, as the environments used in this work contain no more than 10 agents, we assign a distinct mask for each agent. However, when the problem scales to hundreds of agents, this vanilla implementation may fail. In such cases, a possible approach is to cluster $N$ agents into $K$ $(K < N)$ groups and train $K$ masks with Kaleidoscope. This would reduce computational costs and achieve a better trade-off between sample efficiency and diversity. Within the same group, agents share all parameters, while agents from different groups share only partial parameters. Techniques for clustering agents based on experience, as proposed by Christianos et al. (2021), could be useful.

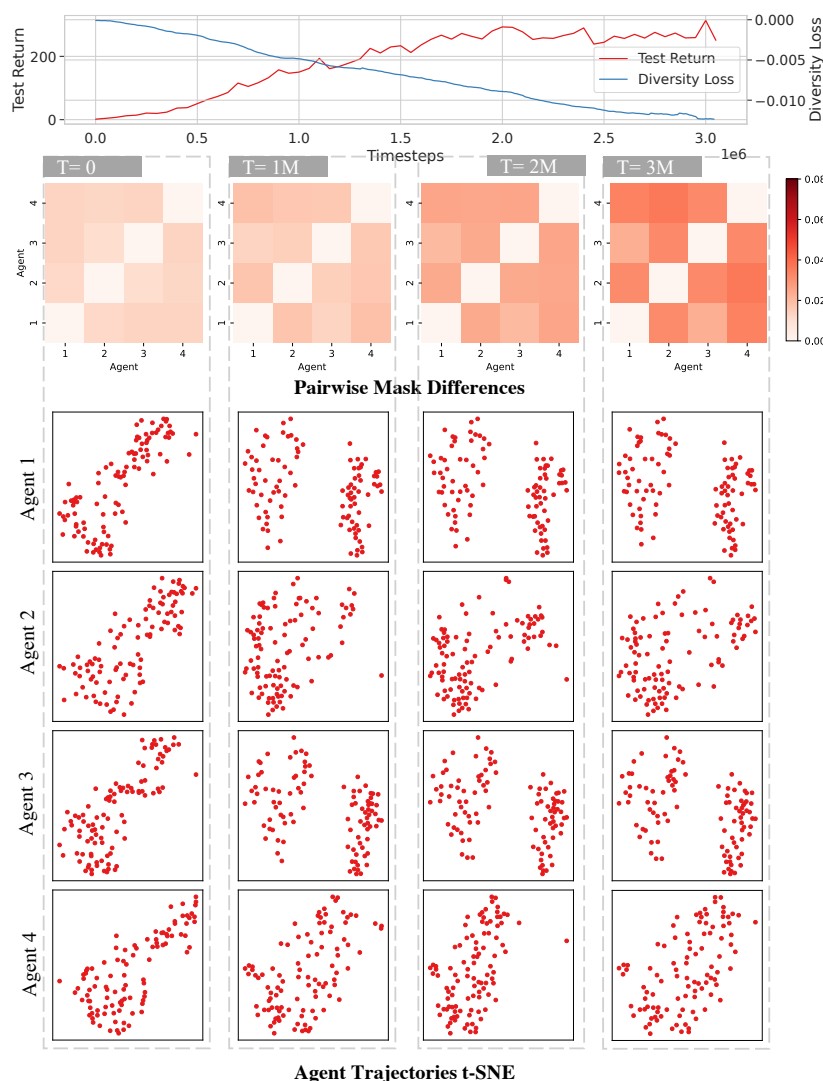

Figure 10: Further visualization on `World`.

