# OpenReview forum: "Kaleidoscope: Learnable Masks for Heterogeneous Multi-agent Reinforcement Learning"
_NeurIPS.cc/2024/Conference — NeurIPS 2024 poster_

### Official Review · Reviewer_CAY2 · 2024-07-08

**Soundness:** 3
**Presentation:** 3
**Contribution:** 2
**Rating:** 6
**Confidence:** 5

**Summary:**

The authors propose a method for adaptive parameter sharing in Multi-Agent Reinforcement Learning (MARL), by using learnable weight masks for each agent. They combine this with a regularization method to encourage diversity in the masks and a resetting mechanism to reuse masked parameters with a certain probability. They showed that their method can outperform parameter sharing baselines on a set of MARL benchmarks.

**Strengths:**

- The paper is well-written and easy to follow. Their methodology and context are explained well. Furthermore, the authors provide a clear motivation for their work, to allow heterogeneous behaviour in MARL, while still leveraging the sample efficiency of parameter sharing, which is an important problem in MARL.
- Applying soft threshold reparameterization (STR) [1] in MARL appears to be novel.
- The authors provide a range of experiments on common MARL benchmarks. They also provide a detailed ablation study to show the importance of the different components of their method and measure the computation cost (FLOPs) of their method during test time.  They also provide a hyperparameter analysis in the appendix, which is useful.

1] Kusupati, A., Ramanujan, V., Somani, R., Wortsman, M., Jain, P., Kakade, S., & Farhadi, A. (2020, November). Soft threshold weight reparameterization for learnable sparsity. In International Conference on Machine Learning (pp. 5544-5555). PMLR.

**Weaknesses:**

- The authors do not include FuPS+ID (full parameter sharing, conditioning on one-hot encoded agent ids) as a baseline in their experiments. This was used as a baseline in prior work [1,2] and is the most common parameter sharing implementation in MARL, therefore it would be a useful comparison to have.
- The authors appear to have missed a relevant related work [3] that also uses pruning masks for adaptive parameter sharing in MARL. This should be included in the related work section and discussed in the context of the proposed method.
- The motivation for using a method like STR, which learns different use different weights for each agent, versus a method that uses different activations per agent (like SNP [2]) is not clear to me. I understand that STR learns the pruning schedule itself and this is useful, but it is not clear that learning weights is the best way. A discussion on this might be relevant if future works build on these methods.
- From the experimental results, it is not clear that the proposed method is significantly better than the baselines. A lot of the results of the proposed method are within 95% confidence intervals of other methods (and vice versa). This makes it hard to say that the proposed method is significantly better. I recommend using the Interquartile Mean (IQM) as suggested by [4], with at least 5 seeds (as you have done for all environments except SMAC V2) to help clarify if the proposed method is better in a statically significant way. Furthermore, I am unable to find the details of the evaluation procedure in the paper, such as the evaluation interval and how many evaluation steps are used. This should be included in the paper.
- An analysis of the computation cost (speed and memory) during training time might be relevant. Parameter sharing generally reduces the number of parameters required, but with learnable masks, this might not be the case. This should be discussed in the paper.

1] Christianos, F., Papoudakis, G., Rahman, M. A., & Albrecht, S. V. (2021, July). Scaling multi-agent reinforcement learning with selective parameter sharing. In International Conference on Machine Learning (pp. 1989-1998). PMLR.

2] Kim, W., & Sung, Y. (2023). Parameter sharing with network pruning for scalable multi-agent deep reinforcement learning. arXiv preprint arXiv:2303.00912.

3] Li, D., Lou, N., Zhang, B., Xu, Z., & Fan, G. (2024, April). Adaptive parameter sharing for multi-agent reinforcement learning. In ICASSP 2024-2024 IEEE International Conference on Acoustics, Speech and Signal Processing (ICASSP) (pp. 6035-6039). IEEE.

4] Agarwal, R., Schwarzer, M., Castro, P. S., Courville, A. C., & Bellemare, M. (2021). Deep reinforcement learning at the edge of the statistical precipice. Advances in neural information processing systems, 34, 29304-29320.

**Questions:**

1. Is there a motivation for using STR over other methods that learn different activations per agent, like SNP?
2. Can you provide more details on the evaluation procedure, such as the evaluation interval and how many evaluation steps are used?
3. Was there a reason for not including FuPS+ID as a baseline in the experiments?

**Limitations:**

The authors address the limitations in the appendix.

---

> ### Author Rebuttal · Authors · 2024-08-07
>
> Thank you for your insightful feedback. We appreciate your questions and suggestions and have provided clarifications below. Please let us know if you have any follow-up questions or comments; we would be happy to discuss further.
>
> **Q1:** The authors do not include FuPS+ID ... a baseline in prior work [1,2] and is the most common ...
>
> **A1:** We apologize for the misunderstanding. **The FuPS in the original manuscript in fact refers to FuPS+ID, as the reviewer mentioned.** We agree that adding agent IDs is the default PS implementation in MARL, and therefore we did not explicitly mention it. We will rename it as FuPS+ID for clarity.
>
> **Q2:** .. missed a relevant related work [2] that also uses pruning masks for adaptive parameter sharing in MARL...
>
> **A2:** We appreciate the suggestion and will incorperate discussion regarding this concurrent work into the related work section: AdaPS [2] combines SNP and SePS by proposing a cluster-based partial parameter sharing scheme. \
> It is also worthnoting that in our manuscript, "adaptive" refers to whether the sharing scheme evolves during training (see Table 1 in our manuscript). Although more flexible than SePS, AdaPS is not adaptive in this sense because its masks are initialized at the beginning of training and kept fixed throughout the training. Without the learnable masks, AdaPS cannot adaptively adjust the parameter sharing scheme during the training to boost the performance. Its performance highly relies on good initialization. A fixed partial parameter sharing scheme in general can not exhance the joint network representational capacity as our proposed Kaleidoscope does (see analysis in line 136-143 in our manuscript).
>
>
> **Q3:** The motivation for using ... different weights for each agent, versus ... different activations ... is not clear to me. I understand that STR learns the pruning schedule itself and this is useful, but it is not clear that learning weights is the best way...
>
> **A3:** We appreciate this insightful question. In fact, we also considered this when designing our method. During the rebuttal period, we conducted an experiment to justify this design choice, with results provided in Fig. 9 in the response PDF. As the results indicate, masking weights is more effective than masking activations. The rationale is that masking weights offers better flexibility. In a fully-connected layer, masking an activation is equivalent to masking all the weights connected to that activation. In Kaleidoscope, masks are learnable. In mask learning, better flexibility corresponds to more parameters to optimize, increasing the chance of finding optimal masks. \
> We agree this is an important design choice to justify and will incorporate these new results and discussions into our manuscript.
>
> To sum up, comparing with baselines, **our Kaleidoscope enjoys better flexibility at two levels**:
> - The learnable nature of masks provides adaptive PS during training (as you already mentioned).
> - The weight-level sharing allows for partial sharing that is more precise (as verified by this additional experiment).
>
>
> **Q4:** ... not clear that the proposed method is significantly better than the baselines... I recommend using IQM ... to help clarify if the proposed method is better in a statically significant way ... details of the evaluation procedure ...
>
> **A4:** Following your suggestion, we now report IQM with 80% CI in the response PDF. For more seeds in SMAC v2, we are currently running those, but the results are not fully ready due to the limited time of the rebuttal period. Regarding the evaluation procedure, we follow the standard configurations provided by the corresponding codebase, listed below for your reference:
>
> | Environment   | Evaluation Interval | Evaluation Steps/Episodes |
> |-----------|---------------------|------------------|
> | MPE       | 50000 steps         | 100 episodes|
> | MaMuJoCo  | 10000  steps        | 40 episodes|
> | SMACv2    | 10000  steps        | 32 episodes|
>
>
> **Q5:** An analysis of the computation cost (speed and memory) during training time might be relevant...
>
> **A5:**
>
> - **Cost results:** We follow your suggestion and summarize the training and execution costs of all methods in Table 2,9,10 in global rebuttal text. **Overall, Kaleidoscope enjoys a higher performance with lower execution complexity but does introduce some extra training overhead.** The training overhead is comparable with some baselines such as SePS.
> - **Discussion:** Kaleidoscope sacrifices training computational cost for the following benefits:
>     - During training, Kaleidoscope achieves better policy diversity while maintaining high sample efficiency.
>     - During execution, Kaleidoscope enjoys better performance with lower overhead.
>
>     **We believe this tradeoff is worthwhile** because:
>     - In a learning task, training is conducted once. After deployment, execution performance and efficiency are what matter.
>     - In an RL setting, sample efficiency is more crucial than computational efficiency because online data in the real world is usually difficult to collect.
>     - Training computational cost can potentially be reduced by employing strategies such as mixed precision training.
>
> We will incorporate these results and discussions into our manuscript.

---

> > ### Author Response · Authors · 2024-08-12
> >
> > To further clarify the issue with agent IDs (Q1&A1) and prevent any confusion, we ran the FuPS baseline (without agent IDs) during the rebuttal period. The results for converged performance (test return for MPE/MaMuJoCo and test win rate for SMACv2) are provided below. Overall, FuPS performs significantly lower than both FuPS+ID and our proposed Kaleidoscope in most scenarios, aligning with the observations made by [1, 3].
> >
> > | Environment   | FuPS     | FuPS+ID | Kaleidoscope|
> > |---------------|----------|---------|-------------|
> > | World              | 209.732 (18.896) | 249.752 (15.367) | **305.374 (42.158)** |
> > | Push               | **12.835 (1.792)** | 9.947 (2.53) | **12.485 (1.997)** |
> > | Ant-v2-4x2         | 1975.317 (191.602) | 4967.296 (701.624) | **6323.875 (290.246)** |
> > | Hopper-v2-3x1      | 285.393 (33.23) | 3074.016 (759.795) | **3218.359 (650.836)** |
> > | Walker2D-v2-2x3    | 4529.468 (679.561) | 5602.741 (688.268) | **6294.386 (1427.098)** |
> > | Walker2D-v2-6x1    | 748.517 (146.915) | 6174.834 (1401.284) | **7311.029 (1861.8)** |
> > | HalfCheetah-v2-2x3 | 5405.805 (355.456) | 6463.89 (2390.081) | **7679.438 (604.483)** |
> > | Swimmer-v2-10x2    | -23.263 (40.216) | 448.524 (14.803) | **474.805 (17.664)** |
> > | Terran_5_vs_5      | 60.26 (9.269) | 59.201 (8.539) | **60.857 (9.201)** |
> > | Protoss_5_vs_5     | 61.179 (8.534) | **63.065 (8.447)** | 62.385 (8.352) |
> > | Zerg_5_vs_5        | 34.616 (9.172) | 35.122 (8.464) | **39.057 (8.386)** |
> >
> > As the rebuttal period comes to a close, we kindly ask if our original rebuttal, along with this additional baseline, has satisfactorily addressed your initial concerns. If so, we respectfully request you to consider updating your review score based on our responses. However, if you have any remaining questions or need further clarification, please do not hesitate to let us know. Thank you.
> >
> >
> >
> > [1] Christianos, Filippos, et al. "Scaling multi-agent reinforcement learning with selective parameter sharing." International Conference on Machine Learning. PMLR, 2021. \
> > [3] Kim, Woojun, and Youngchul Sung. "Parameter sharing with network pruning for scalable multi-agent deep reinforcement learning." arXiv preprint arXiv:2303.00912 (2023).

---

> > > ### Comment · Reviewer_CAY2 · 2024-08-12
> > >
> > > Thanks to the authors for their response. Most of my concerns have been addressed. I raised my score by 2 points.

---

> > > > ### Author Response · Authors · 2024-08-12
> > > >
> > > > Thank you so much for raising the score and for providing useful suggestions in the initial review. We are glad that our response addressed your concerns satisfactorily.

---

### Official Review · Reviewer_BN4p · 2024-07-12

**Soundness:** 3
**Presentation:** 2
**Contribution:** 3
**Rating:** 6
**Confidence:** 4

**Summary:**

This paper presents an approach to multi-agent reinforcement learning.  In this approach, the agent network has an overall set of parameters.  These parameters are transformed by agent-specific masks.  The masks are learnable.  Agent policies are encouraged to be diverse through a diversity-regularisation term.  Masks and diversity regularisations are also applied to the critic network parameters.  Experimental results compare the performance of Kaleidoscope with other methods.

**Strengths:**

The introduction of masking as part of the learnable parameters for MARL agents is interesting.  It removes a previous limitation of parameter sharing, and allows this to be flexible.  The contributions are made clear by the authors. Figure 2 illustrates the concept of masking well, which is important to the core of the work.  The manuscript is generally easy to read, barring some minor comments pointed out in the Questions section.

**Weaknesses:**

- The current limitations of the work are presented in an Appendix.  It would be good to see this fleshed out with more details and presented as part of the main text.  For example, it would be interesting if the authors provide a guideline for how their method would scale with the number of agents.
- While the presentation of the material is clear, it is repetitive in places.  It would be nice to make the text more concise and provide some more interpretation of the results.
- In general, the Figure captions can be made more self-contained.
- Minor comments:
   * In Section 2, after line 78, it appears that the meaning of $O$ is not explained in the MDP $\mathcal{M}$.
   * Line 89 appears to have either grammatical errors or a confusing choice of words between singular and plural.
   * Line 277 has a typo in MARL.

**Questions:**

It would be nice if the authors are able to address the points in the Weaknesses section.

**Limitations:**

No.  The current limitations of the work are presented in an Appendix.  It would be good to see this fleshed out with more details and presented as part of the main text.  For example, it would be interesting if the authors provide a guideline for how their method would scale with the number of agents.

---

> ### Author Rebuttal · Authors · 2024-08-07
>
> Thank you for your positive review. Regarding your questions and suggestions, we would like to provide clarifications below. If you have any follow-up questions or comments, please let us know, and we will be happy to discuss further.
>
> **Q1:** The current limitations of the work are presented in an Appendix. It would be good to ... as part of the main text. For example, it would be interesting if the authors provide a guideline for how their method would scale with the number of agents.
>
> **A1:** We appreciate the suggestion and will elaborate on how to protentially scale Kaleidoscope to more agents. Instead of assigning one unique mask to each agent, we can consider clustering $N$ agents into $K$ ($K \lt N$) groups and train $K$ masks with Kaleidoscope. This would reduce computational costs and achieve a better trade-off between sample efficiency and diversity. Within the same group, agents share all parameters, while agents from different groups share only partial parameters. Techniques for clustering agents based on experience, as proposed in [1], could be useful.
>
> **Q2:** While the presentation of the material is clear, it is repetitive in places. It would be nice to make the text more concise and provide some more interpretation of the results.
>
> **A2:** We appreciate the suggestion and provide further visualization of agent trajectories in Fig. 7 in the response PDF. We visualize the pairwise mask differences among agents and the agent trajectories at different training stages. As training progresses, the test return increases and diversity loss decreases, indicating better performance and greater diversity among agent policies. Correspondingly, mask differences among agents increase, and the agent trajectory distribution becomes more diverse. This reflects:
>
> - The masks become more diverse as the training proceeds.
> - The more diverse masks results in more diverse policies (reflected by the trajectories).
> - The diverse policies lead to better performance.
>
> **Q3:** More self-contained Figure captions and typos.
>
> **A3:** Thank you for catching those, and we will revise them accordingly.

---

> > ### Comment · Reviewer_BN4p · 2024-08-11
> > **Response to Rebuttal by Authors**
> >
> > Thank you for the response, particularly on the scalability of the method.

---

> > > ### Author Response · Authors · 2024-08-12
> > >
> > > We are glad to see that your initial concerns have been addressed. Again, we truly appreciate your time and effort in helping us improve our work.

---

### Official Review · Reviewer_Y5Eh · 2024-07-13

**Soundness:** 2
**Presentation:** 2
**Contribution:** 2
**Rating:** 4
**Confidence:** 5

**Summary:**

The paper introduces a novel adaptive partial parameter sharing scheme in multi-agent reinforcement learning (MARL) to enhance policy heterogeneity while maintaining high sample efficiency. The key innovation, Kaleidoscope, employs a set of common parameters and multiple sets of distinct, learnable masks for different agents, encouraging policy diversity without sacrificing the benefits of parameter sharing. This approach dynamically balances the trade-off between full parameter sharing and non-parameter sharing, thereby bridging the gap between the two. Furthermore, Kaleidoscope is extended to critic ensembles within actor-critic algorithms to improve value estimations. Empirical evaluations across various environments, including multi-agent particle environment, multi-agent MuJoCo, and StarCraft multi-agent challenge v2, demonstrate that Kaleidoscope outperforms existing parameter sharing methods.

**Strengths:**

This paper investigates the limits of parameter sharing in MARL and conducts conprehensive experiments.

**Weaknesses:**

### Methodology
1. The proposed method is similar to SNP, and the difference is not fully discussed.
2. The learned masks can not be explained. The proposed method can be regarded as enlarging the network into a Mix-of-Expert network with a dense router, which is expected to improve the performance. However, the proposed method does not leverage the relationships among agents.
3. Pseudocode is lacking. The proposed method has several components, it is better to include a pseudocode in the main paper.
4. The proposed method maintains N masks whose sizes are similar to those of the actor and critic networks, which may result in low training efficiency.

### Experiments
1. The authors should consider implementing a simple and intuitive partial parameter sharing network as a baseline, such as a shared trunk with multiple action heads.
2. Apart from Table 2, the authors should compare the training costs among those algorithms.
3. The ablation study is incomplete. The main new idea behind the proposed method is to share parameters using learnable masks. Because of this, the authors should show how this main new idea improves performance instead of getting rid of secondary methods in Figure 6.

**Questions:**

1. What is the purpose of citing LLM works in lines 35 and 36?
2. Are there any other works discussing parameter sharing apart from references [12,13,14] in this paper?
3. How to distinguish `Neurons` and `Weights` in Table 1?
4. In the experiments, are algorithms using similar sizes of networks? For a fair comparison, the network sizes for agent policies and critics should be similar.

**Limitations:**

yes

---

> ### Author Rebuttal · Authors · 2024-08-07
>
> Thank you for your constructive feedback. Here are our clarifications. If you have any follow-up questions or comments, please let us know and we will be happy to have further discussions.
>
> **Q1:** The proposed method is similar to SNP...
>
> **A1:** As listed in Table 1, our proposed method differs from SNP in the following key aspects:
> - Kaleidoscope learns masks that indicate which parameters to share during training, whereas SNP relies on fixed mask initialization. We then further design diversity regularization and resetting mechanism to facilitate the mask learning with the objective to boost the task performance.
> - Kaleidoscope applies masks to weights, while SNP applies masks to neurons (activation outputs).
>
> These design choices give **Kaleidoscope greater flexibility** (elaborated below) in parameter sharing and result in better performance (supported by Fig. 4-5).
> - The learnable nature of masks provides adaptive PS during training.
> - The weight-level sharing allows for partial sharing that is more precise (supported by Fig. 9 in the response PDF).
>
> **Q2:** The learned masks can not be explained. The proposed method can be regarded as enlarging the network into a Mix-of-Expert network with a dense router, ... does not leverage the relationships among agents.
>
> **A2:** In Fig. 7 of the original manuscript, we visualize mask differences among agents throughout the training process, reflecting the pairwise similarity between agent policies. Initially, agents share all parameters and gradually learn diverse policies through different masks. During the rebuttal period, we further visualized agent trajectories as training proceeds, with results provided in Fig. 7 of the response PDF.
>
> **Q3:** Pseudocode is lacking...
>
> **A3:** We appreciate the suggestion and will revise the manuscript accordingly. Kaleidoscope generally requires an extra resetting step, as shown in Fig. 3 of the original manuscript. It follows the same end-to-end training procedure as the base algorithms (MATD3 or QMIX), with modified network structures and loss functions.
>
> **Q4:** The proposed method maintains N masks whose sizes are similar to those of the actor and critic networks, which may result in low training efficiency.
>
> **A4:** Please see A6.
>
> **Q5:** ... implementing ... a shared trunk with multiple action heads.
>
> **A5:** We follow the suggestion and implement this baseline as MultiH during the rebuttal period. The results are reported as purple curves in Fig. 3-4 in the response PDF. Overall, the performance of MultiH lies between FuPS and NoPS. Like other non-adaptive partial parameter sharing algorithms, MultiH finds a fixed middle point between full parameter sharing and no parameter sharing. While this particular middle point may be favored in certain scenarios, its performance across different scenarios is not stable due to the lack of flexibility. This observation motivates us to develop an adaptive partial parameter sharing algorithm such as Kaleidoscope.
>
> **Q6:** Apart from Table 2, the authors should compare the training costs among those algorithms.
>
> **A6:**
> - **Cost results:** We summarize the training and execution costs of all methods in Table 2,9,10 in global rebuttal text. **Overall, Kaleidoscope enjoys a higher performance with lower execution complexity but does introduce some extra training overhead.** The training overhead is comparable with some baselines such as SePS.
> - **Discussion:** Kaleidoscope sacrifices training computational cost for the following benefits:
>     - During training, Kaleidoscope achieves better policy diversity while maintaining high sample efficiency.
>     - During execution, Kaleidoscope enjoys better performance with lower overhead.
>
>     **We believe this tradeoff is worthwhile** because:
>     - In a learning task, training is conducted once. After deployment, execution performance and efficiency are what matter.
>     - In an RL setting, sample efficiency is more crucial than computational efficiency because online data in the real world is usually difficult to collect.
>     - Training computational cost can potentially be reduced by employing strategies such as mixed precision training.
>
> **Q7:** The ablation study is incomplete. The main new idea ... share parameters using learnable masks...
>
> **A7:** We appreciate the suggestion and have added an ablation with fixed masks rather than learnable masks in Fig. 6 of the response PDF. This new ablation underperforms Kaleidoscope with the same sparsity level. It showcases the need to use learnable masks to dynamically adjust the parameter sharing throughout the training (as supported by Fig. 7 in the original manuscript).
>
> **Q8:** What is the purpose of citing LLM works in lines 35 and 36?
>
> **A8:** To support the current trend of scaling up model sizes and motivate the need for an effective partial parameter-sharing technique. Although our method focuses on MARL, we believe parameter sharing is potentially a topic of interest in other multi-agent systems. We will polish the wording to make this clearer.
>
> **Q9:** Are there any other works discussing parameter sharing apart from references [12,13,14] in this paper?
>
> **A9:** The references in our manuscript are good representatives of recent progress on parameter sharing in MARL. However, as reviewer BN4p suggested, we will add another concurrent work to the related work section: AdaPS [2], which combines SNP and SePS by proposing a cluster-based partial parameter-sharing scheme.
>
> **Q10:** How to distinguish Neurons and Weights in Table 1?
>
> **A10:** Neurons refer to the output of the activation function, while weights refer to the network parameters. In a fully-connected layer, removing a neuron is equivalent to removing all the weights associated with that neuron.
>
> **Q11:** ... are algorithms using similar sizes of networks?...
>
> **A11:** Yes, we use the same network structure across different algorithms, with details listed in Appendix A.1.3.

---

> > ### Comment · Reviewer_Y5Eh · 2024-08-11
> > **Responses to authors**
> >
> > Thanks to the authors' reply, which addressed most of my concerns. I raised my score by 1 point.

---

> > > ### Author Response · Authors · 2024-08-12
> > >
> > > Thank you so much for raising the score and for providing valuable suggestions in the initial review.
> > >
> > > We are pleased that most of your concerns have been addressed, and we are eager to discuss further. To give you a better overview, we summarize our contributions: In our work, we propose **a novel adaptive partial parameter sharing scheme, Kaleidoscope,** which fosters policy heterogeneity while maintaining high sample efficiency in MARL tasks. This approach leads to **superior policies in terms of both performance and execution cost across a wide range of MARL benchmarks**. Additionally, the flexibility of Kaleidoscope makes it easy to integrate with various MARL algorithm families.
> > >
> > > During the rebuttal period, we included the following experiments to address your concerns:
> > > - More informative visualization (Fig. 7 in the response PDF)
> > > - Justification for using weight-level masking instead of neuron-level masking (Fig. 9 in the response PDF)
> > > - Baseline with multiple action heads (Fig. 3-4 in the response PDF)
> > > - Training costs (Table 2,9,10 in global rebuttal text)
> > > - Ablation study on learnable vs. fixed masks (Fig. 6 of the response PDF)
> > >
> > > We plan to incorporate these into the updated manuscript and further refine the writing to enhance clarity, as per your suggestions. Given the limited space for the initial rebuttal, some answers might need further clarification. If so, please let us know if there are any additional concerns. We truly value your input and would be happy to discuss further.

---

> > > > ### Author Response · Authors · 2024-08-13
> > > >
> > > > Thank you once again for taking the time to review our paper and provide your valuable feedback. As the rebuttal period comes to a close, we wanted to check if you have any remaining questions or need further clarification. Please don't hesitate to let us know.

---

> > > > > ### Comment · Reviewer_Y5Eh · 2024-08-14
> > > > > **I have no preference about accepting or rejecting this article**
> > > > >
> > > > > I appreciate the author's detailed response and experimental supplement, and I also see the author's efforts. The reason why I did not further improve the score is just because the code is not provided, but it does not mean that I am against accepting this article (I have no preference about accepting or rejecting this article).

---

> > > > > > ### Author Response · Authors · 2024-08-14
> > > > > >
> > > > > > Thank you for your reply. As indicated in the paper (line 514), the code will be open-sourced upon publication of the paper. Unfortunately, it is against policy to provide external links or upload files during the discussion phase. Therefore, we have sent our code through an anonymized link to the AC in a separate comment, thereby adhering to the policy. Please be assured that we will openly share the full code implementation after this paper completes the review process. We hope this can address your concerns and again we truly appreciate your time throughout the reviewing process.

---

### Official Review · Reviewer_WQkZ · 2024-07-15

**Soundness:** 4
**Presentation:** 4
**Contribution:** 3
**Rating:** 7
**Confidence:** 5

**Summary:**

This paper introduces Kaleidoscope, an adaptive partial parameter sharing method for multi-agent reinforcement learning (MARL). Kaleidoscope aims to balance the benefits of full parameter sharing (sample efficiency) with the flexibility of non-parameter sharing (policy diversity). It achieves this by using learnable binary masks to control which parameters are shared between agent networks and within critic ensembles. The authors demonstrate Kaleidoscope's effectiveness on various MARL benchmarks, showcasing superior performance compared to existing parameter sharing approaches.

**Strengths:**

*Originality*: The application of STR for dynamic partial parameter sharing in MARL is novel and well-motivated. The paper clearly distinguishes itself from prior work on partial parameter sharing that relies on fixed initializations.

*Quality*:  The proposed method is technically sound. The combination of STR with diversity regularization and periodic resetting is well-reasoned and contributes to the effectiveness of Kaleidoscope. Experimental results across diverse MARL benchmarks provide strong support for the claims. The ablation is also helpful to understand the contribution each component of Kaleidoscope has on final performance.

*Clarity*: The paper is generally well-written and organized. The core idea of using learnable masks is clearly conveyed through figures and explanations. The technical details are adequately provided, allowing for reproducibility.

*Significance*: Addressing the trade-off between sample efficiency and policy diversity is a crucial challenge in MARL. Kaleidoscope's adaptive and dynamic nature makes it a potentially valuable contribution to the field. The improved performance on diverse benchmarks suggests that the proposed method can be broadly applicable.

**Weaknesses:**

The paper primarily focuses on environments with a relatively small number of agents. The scalability of Kaleidoscope to environments with  tens or hundreds of agents. In that case it is not clear if a large number of masks could have a negative performance compared with the parameter sharing baseline.

**Questions:**

How does the computational overhead of Kaleidoscope scale with the number of agents?

Impact of Mask Sparsity: Does the sparsity level of the masks induced by STR vary across different tasks and environments? Are there any insights into how the sparsity level affects the policy diversity and overall performance?

Integration with other MARL Algorithms: The paper demonstrates Kaleidoscope with MATD3 and QMIX. How straightforward would it be to adapt the method to other MARL algorithms such MAPPO? Are there any specific challenges or considerations?

Real-world Applications: Can you envision any specific real-world applications where Kaleidoscope's ability to promote agent heterogeneity would be particularly beneficial?

**Limitations:**

The authors have properly addressed the limitations of their paper.

---

> ### Author Rebuttal · Authors · 2024-08-07
>
> Thank you for your positive review. Regarding you questions and suggestions, we would like to provide clarifications and additional results below. If you have any follow-up questions or comments, please let us know and we will be happy to have further discussions.
>
> **Q1:** The paper primarily focuses on environments with a relatively small number of agents. The scalability of Kaleidoscope to environments with tens or hundreds of agents. In that case it is not clear if a large number of masks could have a negative performance compared with the parameter sharing baseline.
>
> **A1:**
>
> - The benchmark environments we use (MPE, MaMuJoCo, SMACv2) are widely adopted and challenging in MARL. They cover various characteristics (as specified in Table 7 in Appendix A.2): discrete/continuous action spaces, heterogeneous/homogeneous agent types, and varying numbers of agents. This diversity makes them suitable for testing MARL algorithms.
> - We agree that scalability is a long-standing challenge in MARL. We acknowledge that scalability to hundreds of agents is a potential limitation of our approach, which we have discussed in Appendix B.3.
>     - **Protential solution**: To scale Kaleidoscope to hundreds of agents, a possible approach is to cluster $N$ agents into $K$ ($K \lt N$) groups and train $K$ masks with Kaleidoscope. This would reduce computational costs and achieve a better trade-off between sample efficiency and diversity. Within the same group, agents share all parameters, while agents from different groups share only partial parameters. Techniques for clustering agents based on experience, as proposed in [1], could be useful. However, to truly scale current MARL algorithms to hundreds of agents, besides better parameter sharing schemes, we also need consider problems such as severe partiall observation, effective credit assignment and complex coordination among agents, which we will leave for future work.
>
> **Q2:** How does the computational overhead of Kaleidoscope scale with the number of agents?
>
> **A2:** In theory, the computational overhead of Kaleidoscope and all baselines scales linearly with the number of agents. However, thanks to PyTorch's parallel implementation framework, the actual training and execution time scales sublinearly, unless memory becomes a bottleneck.
>
> **Q3:** Impact of Mask Sparsity: Does the sparsity level of the masks induced by STR vary across different tasks and environments? Are there any insights into how the sparsity level affects the policy diversity and overall performance?
>
> **A3:** Generally, the final sparsity levels of the masks are similar across different tasks. Below are the results, followed by a discussion. Since overall sparsity comparison can be difficult to interpret across environments (affected by network structure, RNN layers, etc.), we report the average sparsity of the fully connected layers where Kaleidoscope is applied
>
> | Scenarios          | Final Sparsity    |
> |--------------------|-------------------|
> | World              | 0.329             |
> | Push               | 0.358             |
> | Ant                | 0.340             |
> | Hopperm            | 0.191             |
> | Walker2D           | 0.211             |
> | HalfCheetah        | 0.350             |
> | Swimmer-v2         | 0.337             |
> | Terran             | 0.248             |
> | Protoss            | 0.226             |
> | Zerg               | 0.321             |
>
>
> Kaleidoscope tends to induce higher sparsity levels with a larger number of agents. However, overall sparsity across different environments does not vary significantly, generally converging between 0.2 and 0.3. This is because an overly sparse network may lack sufficient representational capacity for complex policies, especially if the original network is small, while networks that are not sparse enough may fail to induce diversity.
>
> **Q4:** Integration with other MARL Algorithms: The paper demonstrates Kaleidoscope with MATD3 and QMIX. How straightforward would it be to adapt the method to other MARL algorithms such MAPPO? Are there any specific challenges or considerations?
>
> **A4:**  It should be fairly straightforward to adapt Kaleidoscope to other MARL algorithms because it mainly operates at a network level rather than an algorithm level. We used MATD3 and QMIX to illustrate Kaleidoscope because they represent the actor-critic and value-based families, respectively. While incorporating Kaleidoscope into different families of algorithms may vary slightly, implementation within the same family should be similar. To incorporate Kaleidoscope into MAPPO (another actor-critic algorithm), one should follow the Kaleidoscope-MATD3 implementation.
>
> **Q5:** Real-world Applications: Can you envision any specific real-world applications where Kaleidoscope's ability to promote agent heterogeneity would be particularly beneficial?
>
> **A5:** Kaleidoscope would be most useful in scenarios where agents need to share some aspects of their policies but not end up with identical policies. For example, two dexterous hands cutting a steak: one hand uses a fork to hold the steak, while the other uses a knife to cut it. Here, the agents should share the grasping skill but not compete for the same utensil and cause clashes. Since it's challenging to know the exact level of heterogeneity needed for a specific multi-agent task beforehand, a flexible framework like Kaleidoscope can be beneficial.

---

### Author Rebuttal · Authors · 2024-08-07

We thank all reviewers for their insightful comments and valuable feedback. In our work, we propose a novel adaptive partial parameter sharing scheme that fosters policy heterogeneity while maintaining high sample efficiency in MARL tasks. This approach leads to superior policies in terms of performance and execution cost.

We are pleased that reviewers find our method interesting (BN4p), novel and well-motivated (WQkZ, CAY2), with conprehensive experiments (WQkZ, Y5Eh, CAY2) and good writing quality (WQkZ, BN4p, CAY2). In response to reviewers' concerns and suggestions, we provide the following additional results here and in the response PDF:
- **Further visualization results** (Fig. 7 in the response PDF): We visualize the pairwise mask differences among agents and the agent trajectories at different training stages. As training progresses, the test return increases and diversity loss decreases, indicating better performance and greater diversity among agent policies. Correspondingly, mask differences among agents increase, and the agent trajectory distribution becomes more diverse.
- **Baseline MultiH** - Agents share all parameters except for individual action heads (Purple curves in Fig. 3-4 in the response PDF)
- **Ablation study on fixed masks vs. learnable masks** (Grey curves in Fig. 6 in the response PDF)
- **Comparison results on masking on weights/activations** (Fig. 9 in the response PDF)
- **Updated execution cost and training costs** (Table 2, 9, 10 below)


Table 2: Averaged Testing FLOPs

| Methods       | NoPS | FuPS+ID | MultiH | SePS | SNP   | Kaleidoscope  |
|---------------|------|---------|--------|------|-------|---------------|
| **MPE**       | 1.0x | 1.0x | 1.0x | 1.0x | 0.988x | **0.901x**    |
| **MaMuJoCo**  | 1.0x | 1.0x | 1.0x | 1.0x | 0.900x | **0.680x**    |
| **SMACv2**    | 1.0x | 1.0x | 1.0x | 1.0x | 0.988x | **0.890x**    |

Table 9: Averaged Training Wall Time

| Methods   | NoPS  | FuPS+ID | MultiH | SePS  | SNP    | Kaleidoscope |
|-----------|-------|---------|--------|-------|--------|--------------|
| MPE       | 2.315x| **1.0x**| 1.453x | 2.497x| 1.437x | 2.086x       |
| MaMuJoCo  | 1.0x  | 1.0x    | 1.103x | 0.979x| **0.965x** | 1.524x   |
| SMACv2    | 1.813x| 1.0x    | 2.212x | 3.511x| **0.978x** | 1.379x   |

Table 10: Averaged Training Memory

| Methods   | NoPS  | FuPS+ID | MultiH | SePS  | SNP    | Kaleidoscope |
|-----------|-------|---------|--------|-------|--------|--------------|
| MPE       | 1.005x| **1.0x**| **1.0x**| 1.058x| 1.002x | 1.236x       |
| MaMuJoCo  | 1.004x| **1.0x**| **1.0x**| 1.017x| **1.0x**| 1.059x     |
| SMACv2    | 1.360x| **1.0x**| 1.109x | 1.938x| 1.018x | 3.182x      |



References

[1] Christianos, Filippos, et al. "Scaling multi-agent reinforcement learning with selective parameter sharing." International Conference on Machine Learning. PMLR, 2021. \
[2] Li, Dapeng, et al. "Adaptive parameter sharing for multi-agent reinforcement learning." ICASSP 2024-2024 IEEE International Conference on Acoustics, Speech and Signal Processing (ICASSP). IEEE, 2024.

---

### Comment · Area_Chair_KbFU · 2024-08-07

Dear Reviewers,

The author responses have been uploaded. Please carefully review these responses to see if your concerns have been adequately addressed and actively participate in discussions.

It is important that **all reviewers should acknowledge having read the author responses by posting a comment**, irrespective of whether there is any change in your rating.

Thank you for your cooperation.

Best regards, \
Area Chair

---

### Decision · Program_Chairs · 2024-09-25

**Decision:**

Accept (poster)

**Comment:**

This paper presents Kaleidoscope, a novel approach for adaptive parameter sharing in Multi-Agent Reinforcement Learning (MARL) using learnable weight masks. The method incorporates regularization to promote policy diversity and a resetting mechanism for parameter reuse. The experimental results demonstrate that Kaleidoscope outperforms existing baselines on various MARL benchmarks.

The paper is well-written and provides strong empirical support, with thorough experiments and analysis. Although it initially lacked comparisons with certain baselines and did not fully address some related work, the authors have effectively addressed these issues in their rebuttal. They clarified the role of FuPS+ID and provided additional experiments to justify their approach over alternative methods. Lastly, the initial motivation for using STR over methods that learn different activations per agent was not entirely clear. The authors conducted additional experiments to justify their choice, showing that weight-level masking offers superior flexibility and performance.

Overall, the authors have addressed the reviewers' main concerns effectively. The paper's contributions are substantial, offering a valuable advancement in parameter sharing for MARL. Therefore, it is recommended to accept the paper, provided that the revisions discussed in the rebuttal are incorporated into the final version.